

# Assessment of the Webb-Pearman-Leuning Correction Method for Estimating $CO_2$ Flux in a Tropical Coastal Sea

Muhammad Fikri Sigid[1], Yusri Yusup[1,2], Abdulghani Essayah Swesi[1], Haitem M Almdhun[1], and Ehsan Jolous Jamshidi[1]

[1]Environmental Technology, School of Industrial Technology, Universiti Sains Malaysia, USM 11800, Pulau Pinang, Malaysia
[2]Centre for Marine & Coastal Studies (CEMACS), Universiti Sains Malaysia, Pulau Pinang, Malaysia

*Correspondence to*: Yusri Yusup (yusriy@usm.my)

**Abstract.** $CO_2$ fluxes in coastal waters are vital for the global carbon cycle. The Eddy Covariance technique was used with
open-path gas analyzers to estimate $CO_2$ fluxes. However, these analyzers can lead to overestimation due to water vapor and
temperature effects, and the Webb-Pearman-Leuning (WPL) correction method was applied to improve the accuracy of the
estimated $CO_2$ flux. This study investigates the application of the WPL correction method on $CO_2$ flux measurements over a
tropical coastal sea location. The analysis reveals that the $CO_2$ flux in the coastal waters mainly functions as a sink, with the
diel cycle showing smaller flux magnitudes during the day and increased uptake during the night. The application of the
WPL correction can result in sign changes of $CO_2$ flux, indicating a shift from a $CO_2$ sink to a $CO_2$ source. These sign
changes occur frequently, particularly during afternoon hours, and can significantly impact the implications regarding carbon
exchange. The WPL correction parameters, especially those related to temperature and water vapor fluctuations, play crucial
roles in influencing the $CO_2$ flux variations. The decrease in dry air molar density and increased vertical wind speed within
the correction related to water vapor fluctuations are the major reasons for the sign change of the $CO_2$ flux. Based on the
quality flagging of the WPL correction, the non-sign change $CO_2$ fluxes are predominantly considered reliable data, while
most of the sign change fluxes should be specially checked.

## 1 Introduction

Carbon dioxide ($CO_2$) fluxes can be directly estimated using the eddy covariance (EC) technique (Burba et al., 2013). The
EC method is often used by ecosystem researchers because it has the advantage of quantifying mass (e.g., $CO_2$, methane,
water, etc.) and energy (sensible and latent heat) exchanges of expansive areas, such as forests, croplands, and oceans
(Tokoro and Kuwae, 2018; Heimsch et al., 2021; Lokupitiya et al., 2016; Nakai et al., 2008; Chien et al., 2018).

The EC method uses the understanding of the behavior of turbulent eddies and utilizes vertical turbulent exchange principles
to calculate the flux using the covariance of the high-frequency mixing ratio of $CO_2$ or moisture and the vertical velocity
component of the wind (McGowan et al., 2016; Stull, 1988). High-frequency measurements of wind velocity components are



afforded by sonic anemometers, but the measurement of $CO_2$ or moisture ($H_2O$) mixing ratio requires fast-response analyzers. The Infrared Gas Analyzers (IRGA) can be utilized to measure $CO_2$ or $H_2O$ mixing ratios at high frequencies (e.g., 10 or 20 Hz). At high frequencies, the rapid-response analyzer could capture turbulent exchange and be able to satisfy the EC method requirement (Jones and Smith, 1977).


In air-sea $CO_2$ flux measurements, Webb et al. (1980) introduced a correction method of the Webb-Pearman-Leuning (WPL), which accounts for air density influenced by water vapor and latent heat to address the overestimation error in $CO_2$ flux measured by open-path gas analyzers, caused by the effects of water vapor and temperature (Edson et al., 2011; Broecker et al., 1986; Else et al., 2011). The WPL formulation was developed to eliminate the effects of air density

fluctuations on the molar density of $CO_2$ that could occur in open-path systems (Burba et al., 2008; Miller et al., 2010). Variables of temperature, pressure, and molar density are calculated by the formula to produce the corrected $CO_2$ flux from the gas analyzer. The correction is also necessary because fluctuations in temperature and humidity cause fluctuations in trace gas concentrations and can simulate $CO_2$ flux (Jentzsch et al., 2021), and positive or negative vertical wind velocities can correspond to positive or negative corrections in flux (Liebethal and Foken, 2003). Moreover, Massman and Tuovinen

(2006) confirmed the validity of formulating the WPL terms in terms of the dry air density fluctuations that interpret correctly the turbulent exchange flux, signifying the importance of the WPL correction as an approach to accurate and reliable measurements of surface exchange fluxes.

The significance of the WPL correction factor depends on the ratio between the turbulent fluctuation in constituent

concentration and its mean concentration. A smaller ratio in this regard highlights the greater importance of the WPL correction (Webb et al., 1980). According to Liebethal and Foken (2003), the application of the WPL correction for $CO_2$ flux is considered very important among the constituent fluxes. Their research further demonstrated that the correction has a substantial impact on $CO_2$ flux values, leading to changes of 20% to 30%, significantly higher than the effect on latent heat flux, which only amounts to 2% to 3% of the flux.


Nevertheless, the WPL correction could potentially introduce inaccuracies in water vapor and carbon dioxide measurements with an open-path gas analyzer (Jentzsch et al., 2021). The correction applied to the observed flux can be large, which might potentially lead to significant changes in the measured $CO_2$ flux (Mauder et al., 2021). Moreover, applying the WPL correction can significantly alter $CO_2$ fluxes and lead to unrealistic outcomes in flux measurements under certain

circumstances and conditions, especially in relevant cases with small $CO_2$ fluxes on the order of 1 μmol m$^{-2}$ s$^{-1}$ (Jentzsch et al., 2021).

Some researchers reported that the coastal region is a weak carbon source or uptake (Borges et al., 2005). The net $CO_2$ flux measured in northwestern Taiwan was −1.75 ± 0.98 μmol m$^{-2}$ s$^{-1}$, with the diurnal flux influenced by local wind speed.



Similarly, in Todos Santos Bay, Mexico, the $CO_2$ flux was $-1.32 \pm 8.94$ µmol m$^{-2}$ s$^{-1}$ (Gutiérrez-Loza and Ocampo-Torres, 2016). The $CO_2$ flux at Bodega Bay, California, was also a weak source, with $0.39 \pm 1.84$ µmol m$^{-2}$ s$^{-1}$ during the upwelling period and $0.05 \pm 0.79$ µmol m$^{-2}$ s$^{-1}$ during the relaxation period (Ikawa et al., 2013). Despite their importance, there is still notable uncertainty in how to parameterize these fluxes for global climate models, and more observations are necessary to gain a better understanding of the role of coastal seas in the global carbon cycle (Chien et al., 2018; Doney et al., 2009;

Gutiérrez-Loza and Ocampo-Torres, 2016). Additionally, measuring these fluxes using techniques and corrections is challenging because of the high uncertainties introduced during data processing, especially for smaller fluxes (Else et al., 2011; Prytherch et al., 2010). Coastal waters can display high variability in $CO_2$ flux due to various factors, such as water temperature, salinity, and biological activity (Ikawa et al., 2013). Despite this, wind speed plays a critical role in controlling the magnitude of air-sea $CO_2$ exchanges, and low wind speeds can restrict gas transfer, resulting in reduced $CO_2$ fluxes in

some cases (Aalto et al., 2021). The flux over the coast is low compared to fluxes on land (Zhang et al., 2014; He et al., 2015), and low wind speed over the coast can be one of the reasons that limit gas transfer modulation over coastal waters (Gutiérrez-Loza and Ocampo-Torres, 2016).

Accurate measurements of $CO_2$ flux in coastal waters are essential for a comprehensive understanding of global carbon

processes and for ensuring the precision of future carbon source and sequestration projection studies. Moreover, minor errors could have a significant impact on cumulative fluxes, emphasising a cautious and meticulous approach in interpreting these fluxes (Jentzsch et al., 2021). Given that the WPL correction method has been reported to potentially introduce inaccuracies in $CO_2$ flux measurements, especially in cases of small fluxes, it raises a question regarding its application to $CO_2$ flux measurements in coastal waters. Therefore, the objective of this paper is to assess the WPL correction method in measuring

$CO_2$ flux at a tropical coastal water location.

## 2 Materials and Methods

### 2.1 The EC Dataset

This analysis uses the *in-situ* EC data collected from an automated weather station called the "Muka Head Station" in the Centre for Marine and Coastal Studies of Universiti Sains Malaysia. The station is located on the northwestern part of

Penang, Peninsular Malaysia, at 5°28′06″N, 100°12′01″E, as shown in Fig. 1.

The station measures $CO_2$ and $H_2O$ fluxes and bio-meteorological parameters (global radiation, net radiation, seawater temperature, etc.) of a tropical coastal ocean in the Strait of Malacca. The flux is calculated from the 20-Hz data collected by the open-path LI-7500 infrared $CO_2/H_2O$ analyzer (LI-COR, USA) and a sonic anemometer (RM81000, Young, USA)

(Yusup et al., 2018a). The water level at this spot measures 1.5 meters, and the surface below is composed of sand. The



research site is located on a continental shelf that directly links to the Straits of Malacca, and the site is exposed to minimal anthropogenic influence (Yusup et al., 2018; Yusup et al., 2020).

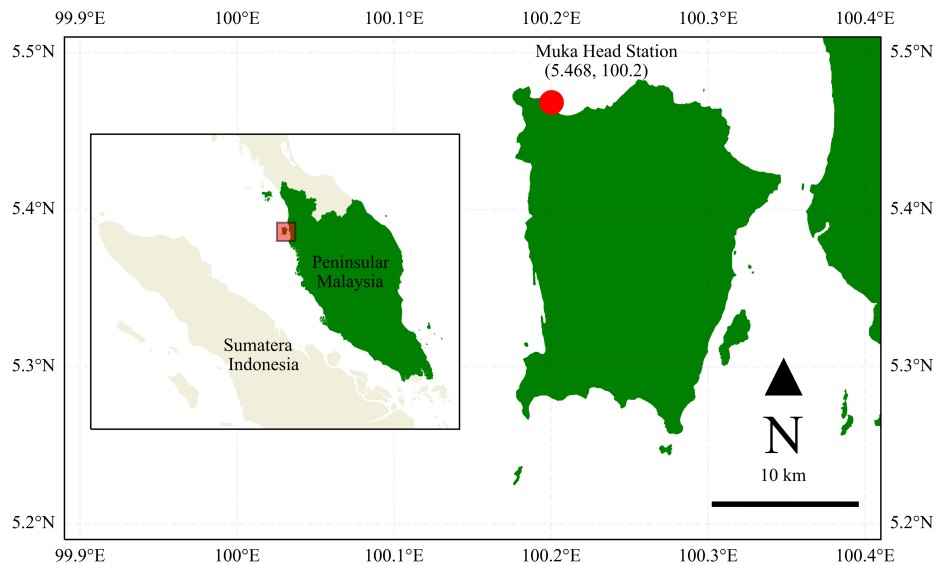

**Figure 1: Red circle and box show the location of the automated weather station called the Muka Head Station in Penang, Peninsular Malaysia.**

From the entire list of variables available in the dataset, the primary variable analyzed was the EC's $CO_2$ flux. The data is accessible at http://atmosfera.usm.my and has a time resolution of 30 minutes. The dataset spanned from 2015 to 2023, but for this analysis, the temporal scope was limited to January 2016 to December 2016 due to the availability of more complete data during the period.

In this research, $CO_2$ fluxes associated with winds originating from directions >315° and <45° were retained, whereas the fluxes with winds coming from other directions were removed during the data processing. This was based on the standard deviation ratio for the vertical wind speed component and the friction velocity, applicable only to wind directions >315° and <90° (Yusup et al., 2018b). Furthermore, wind speed data collected inland from the south to the west of the station (>45° and <315°) were omitted because of the poor-quality flags in the recorded measurements. $CO_2$ flux measurements obtained during rainfall were excluded because of the effects of precipitation on the accuracy of the eddy covariance instruments.




## 2.2 Calculations of the Raw and WPL-Corrected $CO_2$ Fluxes


This study utilized conventional flux calculations that typically rely on density measurements obtained from gas analyzers based on light absorption (Burba et al., 2012). Afterward, density corrections, as described by Webb et al. (1980) and hereafter referred to as WPL, were applied:

$$F_c = \overline{w'\rho_c'} + (1+\mu\sigma)\frac{\overline{\rho_c}}{\overline{T}}\overline{w'T'} + \mu\frac{\overline{\rho_c}}{\overline{\rho_a}}\overline{w'\rho_v'} + (1+\mu\sigma)\frac{\overline{\rho_c}}{\overline{P}}\overline{w'P'} \qquad (1)$$

where $F_c$ is WPL-corrected $CO_2$ flux, $\rho_c$ is molar density of $CO_2$ and $w$ is vertical component of wind speed, $T$ is temperature, $P$ is pressure, $\rho_c$ is molar density of $CO_2$, $\rho_v$ is molar density of water vapor, $\rho_a$ is molar density of dry air (Webb et al., 1980). Meanwhile, $\sigma = \rho_v/\rho_a$ and $\mu = M_a/M_v$ with $M_a$ is molecular weight of dry air, $M_v$ is molecular

weight of water vapor. Overbars represent temporal averages, while primes indicate turbulent deviations from these averages.

The first term on the right-hand side of Eq. (1) is the raw flux covariance, applying the vertical turbulence exchange concept to calculate the flux by using the vertical wind velocity and molar density of $CO_2$. The first term on the equation also refers

as the raw $CO_2$ flux ($F_{c,0}$), namely the $CO_2$ flux that has not been corrected by WPL. Moreover, the WPL formula consists of the corrections for temperature, water vapor, and pressure fluctuations in the open-path gas analyzer, which are stated in the second, third, and fourth terms in Eq. (1). The WPL-corrected $CO_2$ flux ($F_c$) is then analysed and compared to the raw $CO_2$ flux ($F_{c,0}$).

### 2.3 Quality Flagging of the WPL Correction

The quality flagging of the WPL correction ($QF_{WPL}$) was implemented for the quality identification of WPL-corrected $CO_2$ flux, specifically as an approach to mark measurement with immense values of WPL correction, as standard error analyses of eddy-covariance data do not consider the unique nature of potential errors in WPL correction (Jentzsch et al., 2021; Mauder et al., 2013). The $QF_{WPL}$ parameter represents the ratio of the WPL correction to the corrected $CO_2$ flux, calculated using Eq. (2) below:


$$QF_{WPL} = \frac{\mu\frac{\overline{\rho_c}}{\overline{\rho_a}}\overline{w'\rho_v'} + (1+\mu\ \sigma)\ \overline{\rho_c}\frac{\overline{w'T'}}{\overline{T}}}{\overline{w'\rho_c'} + \mu\frac{\overline{\rho_c}}{\overline{\rho_a}}\overline{w'\rho_v'} + (1+\mu\ \sigma)\ \overline{\rho_c}\frac{\overline{w'T'}}{\overline{T}}} \qquad (2)$$

In accordance with general quality flag systems (Foken and Wichura, 1996; Foken et al., 2012), $|QF_{WPL}| \le 0.5$ is categorized as very good, $0.5 < |QF_{WPL}| \le 1$ is classified as good, and any values exceeding $|QF_{WPL}| > 1$ requires a thorough check.





## 3 Results and Discussion

### 3.1 The CO₂ Flux Hourly Cycle at the Tropical Coast

Throughout the sampling time domain, $CO_2$ flux at the study location acted as $CO_2$ uptake, with the average values of $F_{c,0}$ and $F_c$ are –0.14 and –0.0061 µmol m$^{-2}$ s$^{-1}$, respectively. In the diel cycle in Fig. 2, the lowest $CO_2$ flux occurred during the daytime, with the flux closing to equilibrium. The lower flux magnitudes can be attributed to the decrease in wind speed during this period, which lowers the transfer velocity and reduces $CO_2$ flux in accordance with the bulk formula (Wanninkhof, 1992; Wanninkhof et al., 2009). Furthermore, the $CO_2$ flux varied with the hours changing between positive and negative flux, in which $CO_2$ flux during the nighttime displayed greater uptake movements reaching around –0.4 µmol m$^{-2}$ s$^{-1}$ until morning when the flux started tending to be $CO_2$ source to over 0.2 µmol m$^{-2}$ s$^{-1}$.

The $CO_2$ flux during this study is similar to the $CO_2$ flux reported in the Rey–Sánchez et al. (2017) study as carbon uptake, which was conducted at the coastal waters of the Gulf of Aqaba, Israel. Of note is the flux magnitude of this site is substantially lower than the cited study's flux (–1.05 µmol m$^{-2}$ s$^{-1}$). and $F_c$. Notably, it displays pronounced increases and decreases in $CO_2$ fluxes, including instances where the coastal region acts as a source of $CO_2$ during specific periods. Nevertheless, disparities in the temporal patterns are still noticeable. In the reported study, the coast demonstrates $CO_2$ emission (positive fluxes) around 18:00 LT, with negative fluxes starting to decrease from 06:00 LT. In contrast, the fluxes in this study illustrate the coast acting as a $CO_2$ source during the day, with the negative fluxes beginning to decline in the early morning hours.

The standard errors of the mean for the $CO_2$ fluxes are in a range of 0.01 µmol m$^{-2}$ s$^{-1}$ to 0.3 µmol m$^{-2}$ s$^{-1}$. A notable level of uncertainty is found during the morning and evening, but a lower level is observed from 10:00 LT until 19:00 LT when the $CO_2$ flux exhibits less deviation, although sign-changing is more frequent during this period. The observed high uncertainty during specific times may be attributed to fluctuations in evaporation. For instance, an increase in the uncertainty around 08:00 LT was observed due to quite intense fluctuations in vertical wind speed, which may have influenced the $CO_2$ flux's uncertainty.

As shown in Fig. 2c, the sensible heat flux is positive, ranging from 0.5 to 2.5 W m$^{-2}$, with an average of 1.35 W m$^{-2}$. The peak occurred around 08:00 LT and 09:00 LT, during which the positive sensible heat flux exceeded 2 W m$^{-2}$. The magnitude of sensible heat flux can influence the WPL correction absolute value (Jentzsch et al., 2021). Positive (negative) $CO_2$ flux with positive (negative) sensible heat flux would increase $CO_2$ emission (uptake). Meanwhile, in instances of carbon uptake with positive sensible heat flux, the $CO_2$ flux would decrease, and the same holds for emissions in the presence of negative sensible heat fluxes.



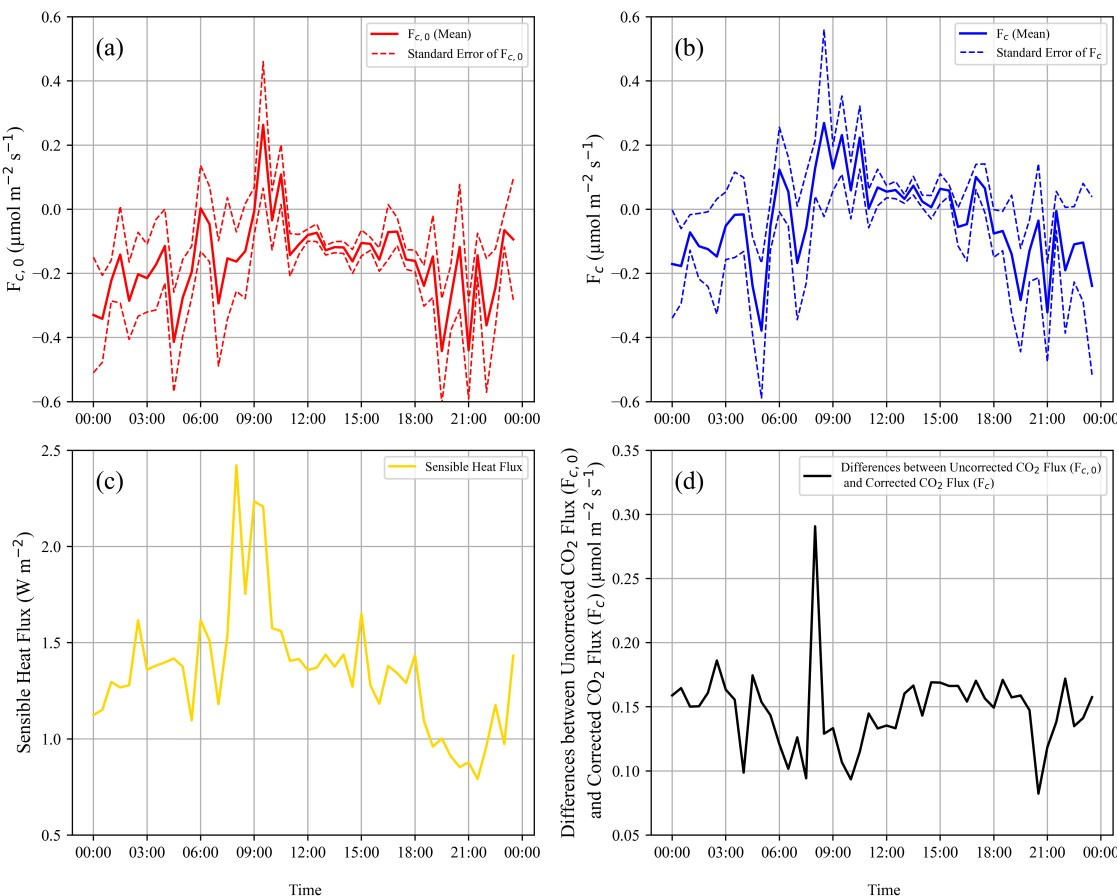

**Figure 2: The climatological variation of diel (a) $F_{c,0}$, (b) $F_c$, (c) sensible heat flux, and (d) difference value between**
180                                    **$F_{c,0}$ and $F_c$ in 2016.**

The difference between $F_c$ and $F_{c,0}$, shown in Fig. 2d, is generally within the range of 0.05–0.2 μmol m$^{-2}$ s$^{-1}$. Distinctively, the difference value at 08:00 LT is 0.29 μmol m$^{-2}$ s$^{-1}$, with a substantial increase of 0.2 μmol m$^{-2}$ s$^{-1}$ from the prior time (07:30 LT) and a noticeable decrease of 0.16 μmol m$^{-2}$ s$^{-1}$ at the following time (08:30 LT). The WPL correction, 185  particularly the third term, could have contributed to the high difference value at 08:00 LT.

The result in difference value between $F_c$ and $F_{c,0}$ is similar to the observation on the open sea by (Kondo and Tsukamoto, 2007), albeit the magnitude difference in this research is not as significant. The magnitude difference by the WPL correction in the cited study is higher by 1.40 μmol m$^{-2}$ s$^{-1}$. The large difference in the latter study was accompanied by a higher 190  average magnitude of the $F_{c,0}$ reaching up to 1.42 μmol m$^{-2}$ s$^{-1}$, which can be due to the location of their study, i.e., the open sea with strong winds. Nevertheless, $F_c$ is much lower than $F_{c,0}$ by over 90% for both over the sea and the coastal waters, and



it results in a $CO_2$ flux value being close to the $CO_2$ flux calculated using the bulk transfer equation as measured in (Kondo and Tsukamoto, 2007).

In addition, the diel cycle between $F_{c,0}$ and $F_c$ shows that $F_c$ has more positive fluxes during the daytime as well as lower negative fluxes due to the WPL correction. In this case, the WPL correction can cause the negative fluxes of the $F_{c,0}$ to change to positive fluxes shown by the $F_c$, which indicates the change in the role of the coast as a sink of $CO_2$ into a $CO_2$ source. The $F_{c,0}$ has the positive fluxes between 09:00 LT and 11:00 LT, whereas the $F_c$ tends to have the positive flux during 05:00 LT until 18:00 LT. Based on research by Jentzsch et al. (2021), the WPL correction applied to $CO_2$ fluxes

smaller than 5 µmol m$^{-2}$ s$^{-1}$ can result in correction values higher than the actual fluxes, which may lead to an uncertain interpretation. Consequently, the different results, particularly in the change of the negative sign to the positive sign of the $CO_2$ flux, can drastically change the conclusion of the carbon exchange in the studied location.

**3.2 Sign Change and Non-Sign Change Occurrences**

During the study period, there were 2689 30-minute data points in 2016. By separating the data into sign change and non-

sign change categories, there were 1237 occurrences of non-sign change, accounting for approximately 46% of the available data. Meanwhile, there were 1452 sign change events, representing around 54% of the available data, slightly higher than the non-sign change occurrences and indicating a significant number of sign change events.

Based on the graph in Fig. 3, sign changes occurred in each 30-minute interval. However, the number of sign change

occurrences, around 10, tended to be lower than the non-sign change occurrences, which were approximately 20 events from evening to morning time. Nevertheless, there were some exceptional times during this period where the number of sign change occurrences exceeded the number of non-sign change occurrences, such as at 02:30 LT (20 sign change occurrences and 16 non-sign change occurrences) and 03:00 LT (22 sign change occurrences and 20 non-sign change occurrences).

The occurrences of both sign change and non-sign change remained relatively steady during the evening time until the morning time. However, the numbers of these two types of data started to increase around 10:00 LT, reaching their peak in the afternoon when the number of sign change occurrences began to surpass the number of non-sign change occurrences. The peak of non-sign change occurred at 12:30 LT with 84 events, gradually decreasing to around 20 events after 15:00 LT. On the other hand, the peak of sign change occurred at 13:30 LT with 113 events before decreasing and returning to their

approximate initial state after 19:00 LT. In the afternoon from 12:00 LT to 19:00 LT, the number of sign change occurrences consistently surpassed the number of non-sign change occurrences, especially after the peak of sign change, where the number of sign change events was twice as high as the number of non-sign change events. Notably, the higher number of both sign change and non-sign change occurrences between 10:00 LT and 19:00 LT indicates that the data from these hours



were not removed as frequently as the data from the rest of the evening to morning time, and in fact, contributed more to the
collective and available utilized data.

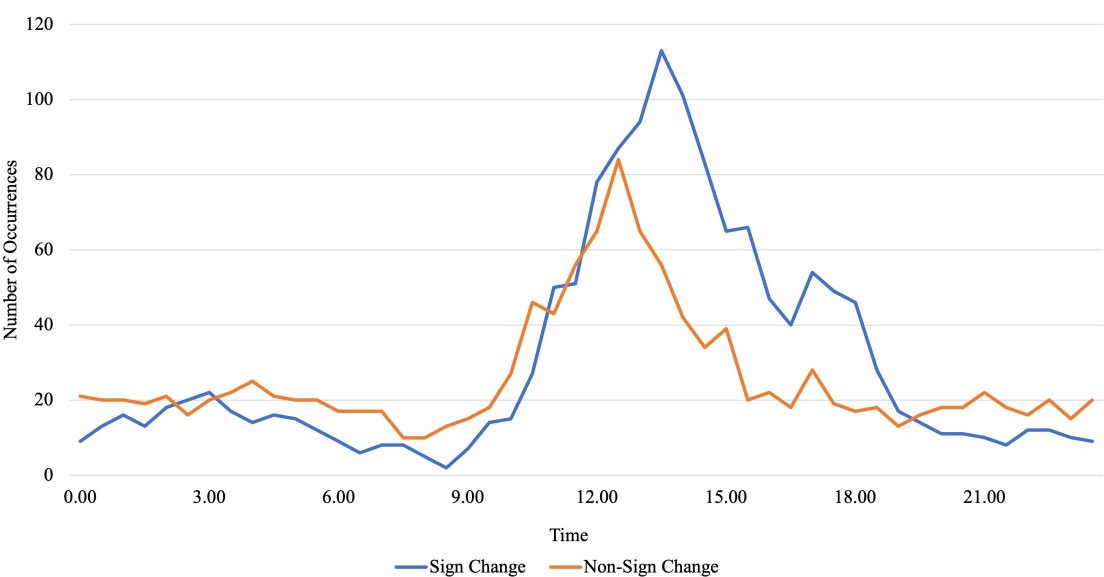

**Figure 3: Quantification of sign change and non-sign change occurrences in a diel cycle.**

The standard error of the mean for each category is lower during the sign change period compared to the non-sign change period. The difference in standard errors might be due to changes in the underlying data distribution during sign change period. During sign change, the data is more centered and tightly distributed. The presence of outliers or extreme values during non-sign change could lead to increased variability and larger standard errors. For $F_{c,0}$ ($F_c$), the average standard errors are 0.0051 (0.0059) µmol m$^{-2}$ s$^{-1}$ during sign change and 0.031 (0.034) µmol m$^{-2}$ s$^{-1}$ during non-sign change. The
corresponding average standard deviation values are 0.084 (0.086) µmol m$^{-2}$ s$^{-1}$ during sign change and 0.82 (0.90) µmol m$^{-2}$ s$^{-1}$ during non-sign change.

Moreover, these values further highlight the lower standard deviation of $CO_2$ flux and the environmental parameters during sign change periods, suggesting more consistent measurements during that time. Conversely, the standard deviation of $CO_2$
flux and those environmental parameters is higher during the non-sign change period. Specifically, during sign change, the standard deviation values for horizontal wind speed, vertical wind speed, temperature, molar density of dry air, and molar density of water vapor are 0.54 m s$^{-1}$, 0.013 m s$^{-1}$, 1.26 K, 0.23 mol m$^{-3}$, and 0.073 mol m$^{-3}$, respectively, while during non-sign change, they are 0.63 m s$^{-1}$, 0.014 m s$^{-1}$, 1.78 K, 0.28 mol m$^{-3}$, and 0.074 mol m$^{-3}$, respectively.



### 3.3 Positive-Negative Sign Change of the $CO_2$ Flux Due to WPL Correction

The sign change typically occurs when the raw $CO_2$ flux is small and close to equilibrium. On average, the mean value of $F_{c,0}$ during the sign change is $-0.11$ µmol m$^{-2}$ s$^{-1}$, whereas the mean value for $F_{c,0}$ during the non-sign change is $-0.169$ µmol m$^{-2}$ s$^{-1}$. In comparison, the mean value of $F_c$ during the sign change is $0.074$ µmol m$^{-2}$ s$^{-1}$, while during the non-sign change it is $-0.096$ µmol m$^{-2}$ s$^{-1}$.

Based on the mean differences calculated as $(F_{c,0} - F_c)/F_{c,0}$, the overall sign change $CO_2$ fluxes are significantly different from zero (p-value < 0.01), including for lower $CO_2$ flux values near zero, specifically below 0.05 µmol m$^{-2}$ s$^{-1}$. The differences for overall sign change $CO_2$ fluxes were found to be $2.47 \pm 3.56\%$ (mean ± 95% confidence interval), and for sign change $CO_2$ fluxes with magnitude below 0.05 µmol m$^{-2}$ s$^{-1}$, they were $3.47 \pm 2.53\%$. Furthermore, the t-test results consistently demonstrated that $CO_2$ flux values during sign change events were significantly different (p-value < 0.01) from
those observed during non-sign change events, even within flux values close to zero. These findings highlight the statistical evidence for the significance of sign change phenomena in this research.

During the study period, the cumulative value of the second term, the third term, and the fourth term in the WPL correction has an average value of 0.15 µmol m$^{-2}$ s$^{-1}$. Additionally, the accumulation of these terms in the WPL correction
demonstrates a higher value during the sign change period compared to the non-sign change period, with average values of 0.187 µmol m$^{-2}$ s$^{-1}$ and 0.105 µmol m$^{-2}$ s$^{-1}$, respectively.

Among these three terms, the two largest correction values are attributed to temperature and water vapor fluctuations, which are represented by the second term and the third term, respectively. The third term of the WPL correction is the primary
component and the most influential factors in altering the sign of $CO_2$ flux when applying the WPL correction. In contrast, the correction for pressure fluctuations, the fourth term, has the least significant effect. The average values for the second term, the third term, and the fourth term of the WPL correction are $-0.0141$ µmol m$^{-2}$ s$^{-1}$, 0.168 µmol m$^{-2}$ s$^{-1}$, $-0.00336$ µmol m$^{-2}$ s$^{-1}$, respectively.

The WPL formula consists of several parameters, including vertical wind speed, temperature, molar density of water vapor, molar density of dry air, and pressure. The average values for each parameter are 0.019 m s$^{-1}$, 302.41 K, 1.24 mol m$^{-3}$, 38.79 mol m$^{-3}$, and 100.74 kPa, respectively. Each term in the WPL correction is determined by a different parameter, with the common factor being the vertical wind speed in each term. The vertical wind speed is the main parameter in the WPL formula.






According to Eq. (1), a higher (lower) vertical wind speed can result in a higher (lower) value for the WPL correction. Notably, the average value of vertical wind speed during the sign change period is $0.0211$ m s$^{-1}$, which is higher than during the non-sign change period, which has an average of $0.0175$ m s$^{-1}$. The results of the t-test also indicate a significant difference (p-value < 0.01) in mean vertical wind speed values between the groups categorized by sign changes and those

without sign changes. Therefore, vertical wind speed can be one of the factors contributing to the sign change of the $CO_2$ flux.

The vertical wind speed can be influenced by the horizontal wind speed since the two are interconnected. During the study period, the average value of horizontal wind speed is $0.92$ m s$^{-1}$. Similar to the vertical wind speed, the horizontal wind

speed during the sign change period (average of $1.05$ m s$^{-1}$) is higher compared to the non-sign change period (average of $0.77$ m s$^{-1}$). The t-test analysis also reveals a significant difference in mean horizontal wind speed values between the sign change and non-sign change groups, indicating statistical significance (p-value < 0.01).

### 3.4 The Correction for Temperature Fluctuations of WPL on $CO_2$ Flux Sign Change

The correction for temperature fluctuations in the WPL formula, specifically the second term of the WPL correction, is

determined by temperature and vertical wind speed. These two factors play a crucial role in determining the second term of the WPL correction value, which can influence the sign of the $CO_2$ flux. The second term is directly proportional to the vertical wind speed but inversely proportional to the temperature. A higher (lower) temperature results in a lower (higher) value for the second term of the WPL correction, while a higher (lower) vertical wind speed leads to a higher (lower) value for the second term of the WPL correction.


During sign changes, the average value from the second term of the WPL correction is $-0.00678$ µmol m$^{-2}$ s$^{-1}$, which is higher than the average value during non-sign changes, which is $-0.0224$ µmol m$^{-2}$ s$^{-1}$. The t-test results also indicate a significant difference (p-value < 0.01) in the mean values when comparing the sign change group to the non-sign change group. This suggests that the average value from the second term of the WPL correction during sign changes has a greater

influence on changing the sign of the $CO_2$ flux from negative to positive.

The average temperature values during sign changes and non-sign changes are $302.65$ K and $302.02$ K, respectively. According to a t-test (with a p-value less than 0.01), there is a significant distinction in mean temperature values between the groups that experienced sign changes and those that did not. The higher (lower) temperature during the sign change (non-

sign change) period can be the main reason for the lower (higher) absolute value of the second term WPL correction.

Additionally, horizontal wind speed may impact temperature. During sign changes, there is a statistically significant negative correlation between horizontal wind speed and temperature (Pearson correlation coefficient, r = $-0.25$). As horizontal wind



speed increases (or decreases), temperature decreases (or increases), which can suggest a cooling effect on the surface air
temperature. In this scenario, a higher horizontal wind speed can lead to a higher vertical wind speed, resulting in a higher
value for the second term of the WPL correction. Furthermore, the lower temperature caused by higher horizontal wind
speed can contribute to a higher value for the second term of the WPL correction.

In contrast to the sign change period, the correlation between horizontal wind speed and temperature during the non-sign
change is statistically significant and positive (r = 0.11). This means that as horizontal wind speed increases (or decreases),
temperature also increases (or decreases) during the non-sign change. The lower horizontal wind speed during the non-sign
change may have a lesser effect on decreasing temperature, leading to the positive correlation between horizontal wind speed
and temperature.

**3.5 The Correction for Water Vapor Fluctuations of WPL on $CO_2$ Flux Sign Change**

The correction for water vapor fluctuations in the WPL formula (the third term of the WPL correction) is determined by the
molar density of $H_2O$, the molar density of dry air, and the vertical wind speed. The third term is directly proportional to the
vertical wind speed but inversely proportional to the molar density of dry air. Lower molar density of dry air and higher
vertical wind speed can result in a higher value for the third term of the WPL correction, potentially causing a sign change in
the $CO_2$ flux. The average value of the third term of the WPL correction during sign changes is 0.196 $\mu$mol m$^{-2}$ s$^{-1}$, which is
significantly higher than during non-sign changes (0.136 $\mu$mol m$^{-2}$ s$^{-1}$; p-value < 0.01).

The Ideal Gas Law and Dalton's law related to the behavior of gas mixtures, specifically the partial pressures and molar
densities of water vapor and dry air, demonstrate the inverse relationship between molar density of $H_2O$ and molar density of
dry air (Miller et al., 2010; Ahrens, 2015). A higher (lower) molar density of $H_2O$ can lead to a lower (higher) molar density
of dry air due to the corresponding increase (decrease) in the partial pressure of water vapor and the decrease (increase) in
the partial pressure of dry air.

Temperature may play a role in the relationship between the molar density of $H_2O$ and the molar density of dry air, as
temperature has an inverse relationship with the molar density of $H_2O$ due to the ideal gas law, where higher temperatures
result in increased kinetic energy, causing gas molecules to occupy larger volumes and decreasing molar density (Andrews,
2010; Atkins and De Paula, 2006). The average molar density of $H_2O$ is lower during sign changes (1.24 mol m$^{-3}$) compared
to non-sign changes (1.25 mol m$^{-3}$), which is opposite to the trend observed in temperature. The mean molar density of $H_2O$
also exhibits a significant difference between the sign change group and the non-sign change group, as indicated by the t-test
(p-value < 0.01).






The formulation of the molar density of dry air demonstrates an inverse relationship with temperature, where a higher (lower) temperature can result in a lower (higher) partial pressure of dry air. In contrast to temperature, but consistent with the molar density of $H_2O$, the molar density of dry air exhibits significant differences during sign changes (average of 38.7 mol m$^{-3}$) compared to non-sign changes (average of 38.8 mol m$^{-3}$; p-value < 0.01). Consequently, higher temperatures can
further increase the value of the third term in the WPL formula by decreasing the molar density of dry air.

Horizontal wind speed and temperature can influence the molar density of water vapor and dry air during sign change periods. In this period, a negative correlation exists between horizontal wind speed and temperature. Higher wind speed can cause lower temperatures, which, in turn, can result in higher molar density of dry air.


Horizontal wind speed can influence the molar density of water vapor through the advection process by causing the horizontal movement of air masses with varying water vapor content. A statistically significant negative correlation (r = – 0.15) was observed during sign changes, indicating higher (lower) horizontal wind speed associated with lower (higher) water vapor molar density. Notably, horizontal wind speed is higher during sign changes compared to non-sign change
periods, in contrast to water vapor molar density. This wind speed influence on water vapor density subsequently can affect the molar density of dry air, leading to a positive correlation between wind speed and dry air density during sign changes (r = 0.17; p-value < 0.01).

During the non-sign change period, a statistically significant negative correlation persists between horizontal wind speed and
the molar density of water vapor (r = –0.26). However, the correlation between horizontal wind speed and the molar density of dry air is weak and not statistically significant (r = 0.05). The absence of a statistically significant correlation between wind speed and dry air molar density in this period could be attributed to a stronger influence of temperature on dry air molar density.

Atmospheric pressure also plays a role in determining the molar density of dry air (Andrews, 2010; Atkins and De Paula, 2006; Ahrens, 2015). Higher (lower) pressure can lead to a higher (lower) molar density of dry air. Notably, the sign change period average pressure (100.73 kPa) is significantly lower compared to the non-sign change period (100.75 kPa; p-value < 0.01). A lower pressure can cause a decrease in the molar density of dry air, subsequently increasing the value of the third term in the WPL formula.

**3.6 Diel Cycle Analysis of WPL Correction Parameters and Their Relationships**

The diel cycle presented in Fig. 4 provides more insights into the parameters involved in the WPL correction. Initially, the diel cycle reveals that vertical wind speed reaches its lowest values, less than 0.02 m s$^{-1}$, around noon, while its peak, over



0.028 m s$^{-1}$, occurs around 03:00 LT and 22:00 LT. The vertical wind speed decreases from around 03:00 LT until approximately noon and then starts to increase in the afternoon until around 22:00 LT.


The diel cycle pattern of vertical wind speed is similar to that of the second term and the fourth term of the WPL correction, showing a direct proportionality to these two terms. The diel cycle of the second term of the WPL correction fluctuates within a range of –0.07 μmol m$^{-2}$ s$^{-1}$ to 0.06 μmol m$^{-2}$ s$^{-1}$. The lower values are observed during the daytime, particularly negative values between 08:00 LT and 19:00 LT, while the rest of the time corresponds to positive values of the second term

WPL correction. On the other hand, the fourth term tends to produce negative values, especially during the daytime, where the values can drop to less than –0.006 μmol m$^{-2}$ s$^{-1}$. From 19:00 LT until 08:00 LT, the values for the fourth term WPL correction are around 0 μmol m$^{-2}$ s$^{-1}$, and they decrease until around 10:00 LT to less than –0.007 μmol m$^{-2}$ s$^{-1}$ before starting to increase again from around 16:00 LT until 20:00 LT.

The diel cycle pattern of vertical wind speed also aligns with that of horizontal wind speed, which ranges from 0.7 m s$^{-1}$ to 1.3 m s$^{-1}$, but exhibits the opposite trend compared to temperature. Temperature reaches its highest values during the daytime, increasing in the morning after 06:00 LT and decreasing after 18:00 LT. This emphasizes the direct proportionality between vertical wind speed and the second term WPL correction, as well as the inverse relationship between temperature and the second term WPL correction in the formula.





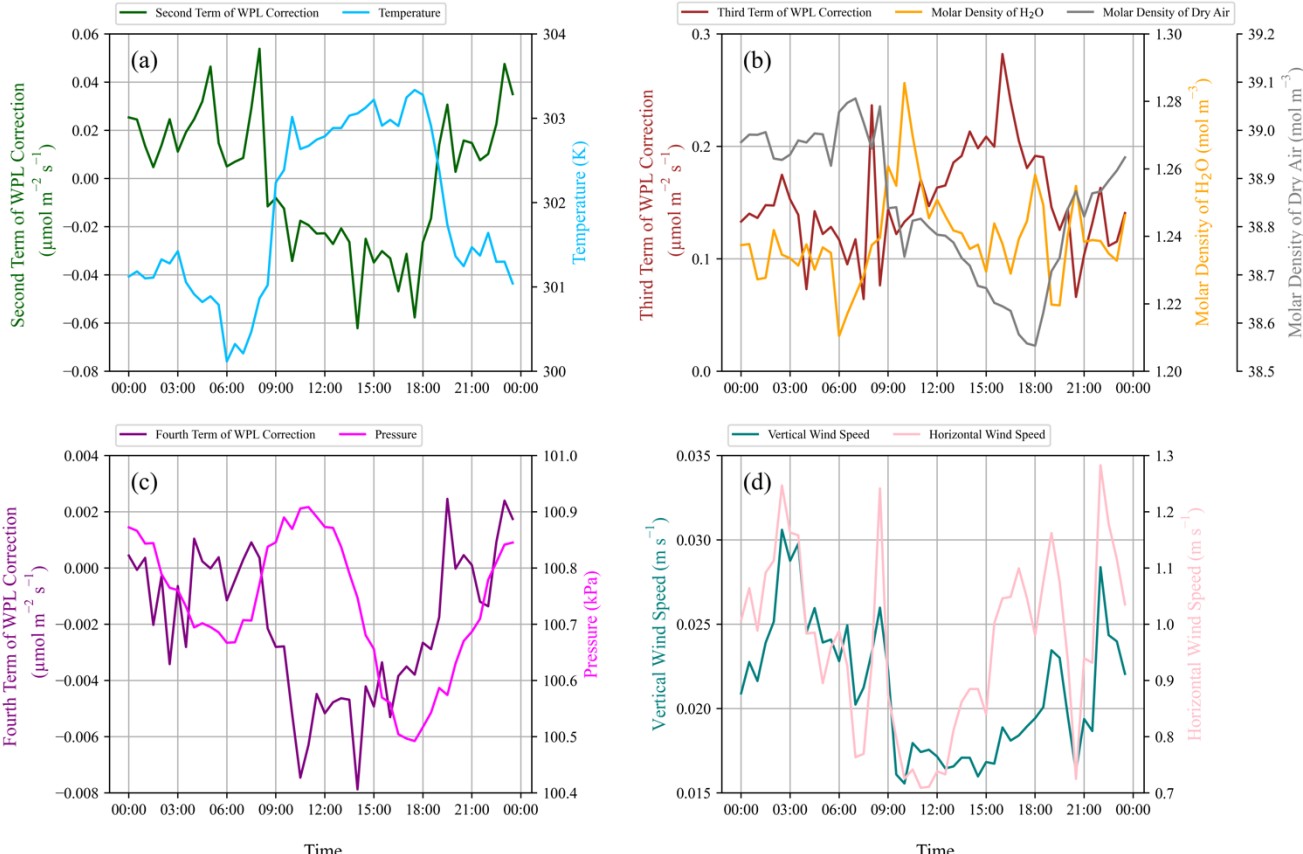

**Figure 4: The climatological variation of diel atmospheric parameters in 2016: (a) Second term of WPL correction and temperature, (b) Third term of WPL correction, molar density of H₂O, and molar density of dry air, (c) Fourth term of WPL correction and pressure, and (d) horizontal wind speed and vertical wind speed.**


The fourth term of the WPL correction follows a similar trend to the vertical wind speed; however, the influence of pressure on the fourth term of the WPL correction is observed between 03:00 LT and 18:00 LT, exhibiting an inverse relationship with pressure during these hours. During the early morning hours, the pressure experiences a slight decrease from approximately 100.87 kPa to less than 100.7 kPa. As the day progresses, the pressure gradually increases, reaching over

100.9 kPa around 11:00 LT. After reaching its peak, the pressure decreases to around 100.5 kPa in the afternoon and then increases again before 18:00 LT, surpassing 100.8 kPa at around 10:00 LT.

Unlike the second term and fourth term of the WPL correction, the third term always has positive values throughout the diel cycle, with higher values during the day. The third term increases before 09:00 LT and reaches a peak of over 0.2 $\mu$mol m$^{-2}$

s$^{-1}$ around 16:00 LT before decreasing to below 0.1 $\mu$mol m$^{-2}$ s$^{-1}$ before 21:00 LT. The diel trend of the third term tends to



be opposite to diel trends of the molar density of $H_2O$ and molar density of dry air, especially during the daytime. It seems that the lower molar density of dry air leads to higher values of the third term WPL correction, contributing to the sign change of the $CO_2$ flux. Based on Fig. 3, sign change occurrences are more frequent after 11:00 LT until 19:00 LT, when the third term of the WPL correction is higher and the molar density of dry air is lower. Temperature might also contribute to the

molar density of dry air during this time, as higher temperatures during the day can lead to a lower molar density of dry air. This indicates that the third term of the WPL correction and molar density of dry air are the main reasons for the sign change of the $CO_2$ flux during the daytime, considering that the second term, fourth term, and vertical wind speed are lower during this time.

During times other than between 11:00 LT and 19:00 LT, when the number of sign change occurrences is lower, the molar density of dry air is not as low as between 11:00 LT and 19:00 LT, but the value of the third term of the WPL correction remains positive, around 0.1 $\mu$mol m$^{-2}$ s$^{-1}$. This suggests that the third term of the WPL correction can still contribute to the sign change of the $CO_2$ flux during times other than between 11:00 LT and 19:00 LT. In contrast to the period between 11:00 LT and 19:00 LT, vertical wind speed is higher during these times, which also leads to higher values of the second term and

fourth term of the WPL correction, and the temperature is lower, potentially further increasing the second term of the WPL correction within the range of 0 $\mu$mol m$^{-2}$ s$^{-1}$ and 0.06 $\mu$mol m$^{-2}$ s$^{-1}$.

### 3.7 $CO_2$ Flux Data with Quality Flagging of the WPL Correction

Based on $QF_{WPL}$, there are a total of 691 data (24.92%) for $|QF_{WPL}| \leq 0.5$ ($QF_{WPL,1}$), 334 data (12.04%) for $0.5 < |QF_{WPL}| \leq 1$ ($QF_{WPL,2}$), and 1748 data (63.04%) for $|QF_{WPL}| > 1$ ($QF_{WPL,3}$). As shown in Fig. 5, fluctuations are evident in all three

variables throughout the day. The number of $CO_2$ fluxes with $QF_{WPL,1}$ varies, with a minimum of 4 and a maximum of 34. $CO_2$ fluxes with $QF_{WPL,2}$ range from 0 to 31, while $CO_2$ fluxes with $QF_{WPL,3}$ range from 9 to 131. Noticeable patterns emerge in the data, with certain periods of the day displaying higher values in all three variables.



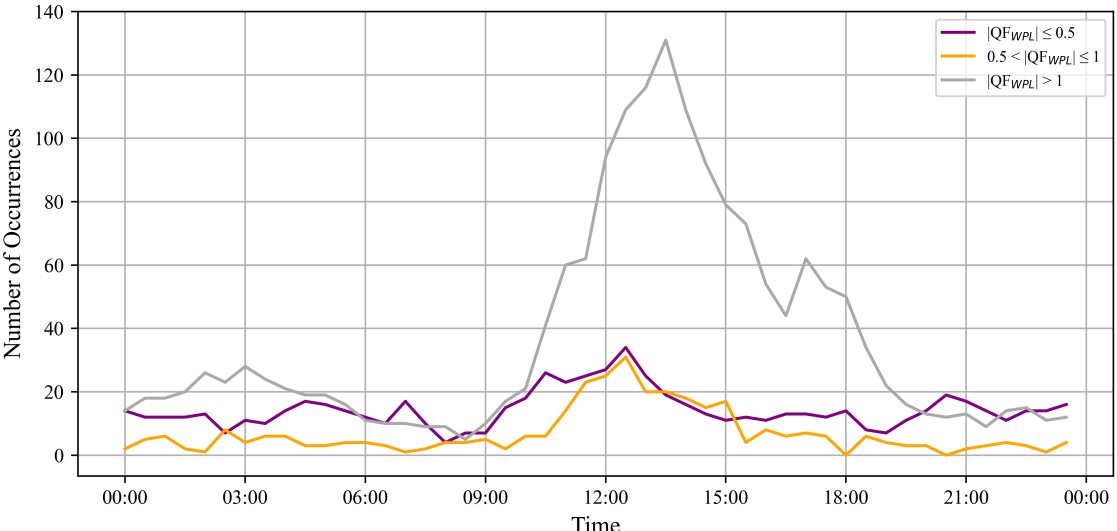

**Figure 5: The QF$_{WPL}$ category occurrences throughout a diel cycle.**

The difference value between QF$_{WPL,3}$ and both QF$_{WPL,1}$ and QF$_{WPL,2}$ is more significant during the daytime than at night, especially at the peak of QF$_{WPL,3}$, although QF$_{WPL,3}$ also surpasses both QF$_{WPL,1}$ and QF$_{WPL,2}$ between 01:30 LT and 04:00 LT. The frequency of occurrences for QF$_{WPL,1}$ and QF$_{WPL,2}$ begins to increase at 08:00 LT (4 instances), peaking at 12:30 LT (more than 30 occurrences). Meanwhile, the number for QF$_{WPL,3}$ starts increasing at 08:30 LT (5 events) and peaks at 13:30 LT (131 events), subsequently dropping to below 20 occurrences after 19:00 LT. These significant QF$_{WPL,3}$ occurrences during the day coincide with higher occurrences of sign change $CO_2$ flux.

Fig. 6a shows the numbers of $F_c$ within |QF$_{WPL}$| ≤ 1, excluding $F_c$ with |QF$_{WPL}$| > 1 and representing WPL-corrected $CO_2$ fluxes that are considered good and very good based on the quality flagging of the WPL correction. Within |QF$_{WPL}$| ≤ 1, 98.38% of the data are non-sign change fluxes. Based on the figure, $CO_2$ fluxes within |QF$_{WPL}$| ≤ 1 are predominantly non-sign change fluxes. Nevertheless, there are still some sign change $CO_2$ fluxes observed between 11:30 LT and 18:30 LT, which can suggest that not all sign change fluxes are errors.

Additionally, within |QF$_{WPL}$| > 1, 83.37% of the data consists of sign change fluxes, while 16.63% represents non-sign change fluxes. Figure 6b, displaying $F_c$ values within |QF$_{WPL}$| > 1, reveals a predominance of sign change $CO_2$ fluxes, especially during the daytime. Although the number of sign change $CO_2$ fluxes is consistently higher than the non-sign change fluxes throughout the period characterized by |QF$_{WPL}$| > 1, it is noteworthy that non-sign change fluxes are still observed. Similar to the sign change $CO_2$ fluxes, the peak of non-sign change $CO_2$ fluxes also occurs in the afternoon.



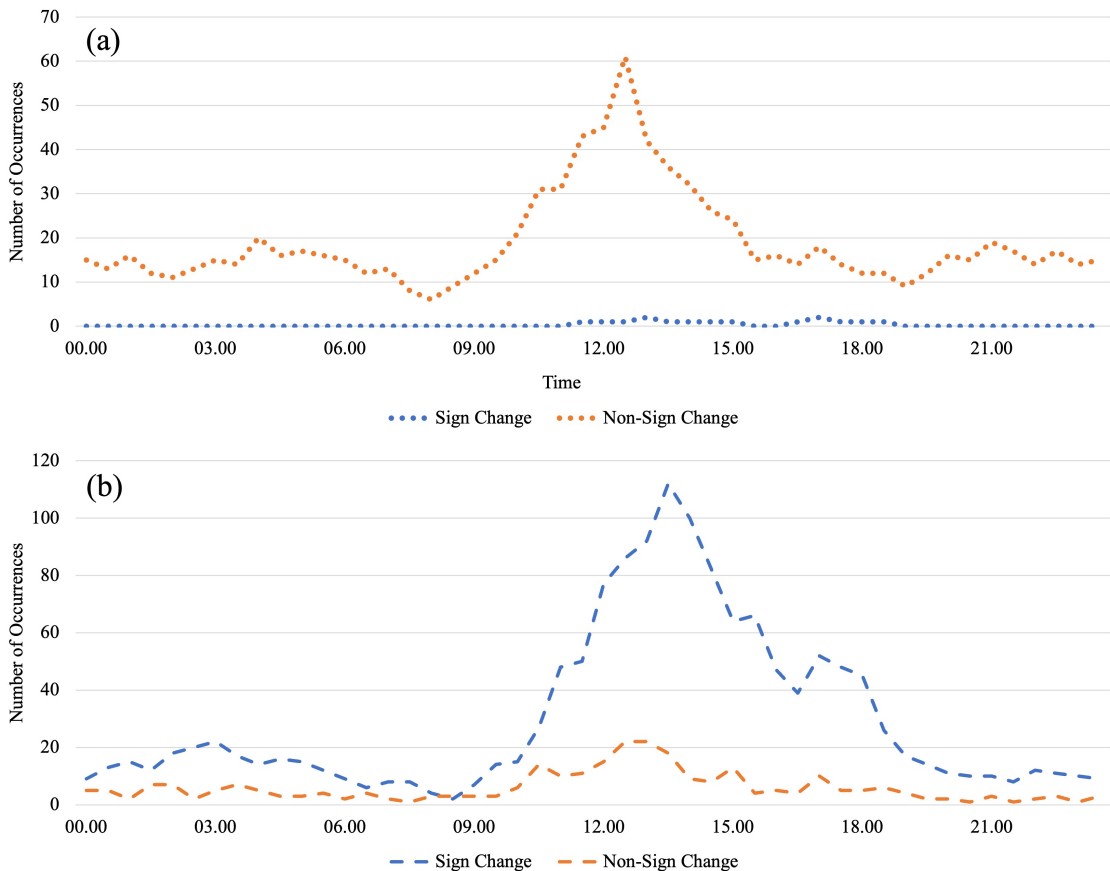

**Figure 6: Quantification of sign change and non-sign change occurrences in a diel cycle within (a) |QF$_{WPL}$| ≤ 1 and (b) |QF$_{WPL}$| > 1.**

As the CO$_2$ flux for |QF$_{WPL}$| ≤ 1 tends to exclude sign change fluxes, the CO$_2$ flux within |QF$_{WPL}$| ≤ 1 exhibits more negative CO$_2$ flux than the fluxes prior to the QF$_{WPL}$ implementation, as illustrated in Fig. 7. Additionally, positive CO$_2$ fluxes are still visible during the day for both $F_{c,0}$ and $F_c$ within |QF$_{WPL}$| ≤ 1. In comparison, $F_c$ during |QF$_{WPL}$| ≤ 1 demonstrates more pronounced positive and negative CO$_2$ fluxes.

In the case of |QF$_{WPL}$| ≤ 1, the average values of $F_{c,0}$ and $F_c$ were –0.185 µmol m$^{-2}$ s$^{-1}$ and –0.155 µmol m$^{-2}$ s$^{-1}$, respectively, both of which represent stronger negative fluxes compared to the fluxes prior to the QF$_{WPL}$ implementation (–0.16 µmol m$^{-2}$ s$^{-1}$ and –0.037 µmol m$^{-2}$ s$^{-1}$, respectively). By excluding $F_c$ with |QF$_{WPL}$| > 1, the difference between $F_c$ and $F_{c,0}$ within |QF$_{WPL}$| ≤ 1 decreases over time, narrowing to a range of 0–0.15 µmol m$^{-2}$ s$^{-1}$. Prior to implementing QF$_{WPL}$, there was a noticeable increase in the difference at 08:00 LT compared to other times. Compared to the fluxes before implementing



$QF_{WPL}$, the difference between $F_c$ and $F_{c,0}$ within $|QF_{WPL}| \leq 1$ at 08:00 LT decreased by 0.188 μmol m$^{-2}$ s$^{-1}$ to 0.1 μmol m$^{-2}$ s$^{-1}$ (35% of the fluxes before $QF_{WPL}$ implementation).

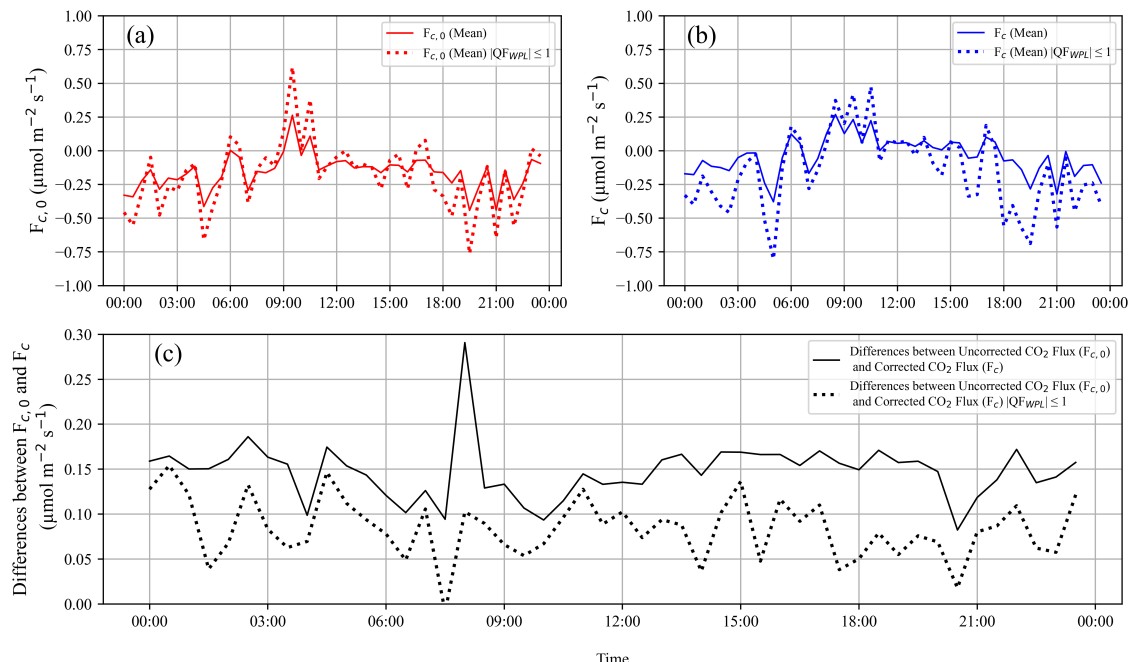

**Figure 7: The climatological variation of diel (a) $F_{c,0}$, (b) $F_c$, and (c) difference value between $F_{c,0}$ and $F_c$, with the**

470                                   **corresponding variables within $|QF_{WPL}| \leq 1$.**

The decrease in the difference between $F_c$ and $F_{c,0}$ can be attributed to the lower absolute values of vertical wind speed, as well as the second and third terms of the WPL correction. At 08:00 LT, the average values within $|QF_{WPL}| \leq 1$ were as follows: vertical wind speed, 0.012 m s$^{-1}$; second term correction, 0.0016 μmol m$^{-2}$ s$^{-1}$; and third term correction, 0.1 μmol

m$^{-2}$ s$^{-1}$. These values represented 51%, 3%, and 42% of the respective values before the $QF_{WPL}$ implementation, and they were also lower by 0.011 m s$^{-1}$, 0.052 μmol m$^{-2}$ s$^{-1}$, and 0.14 μmol m$^{-2}$ s$^{-1}$, respectively. Additionally, the $CO_2$ flux at 08:00 LT also decreased. Prior to the $QF_{WPL}$ implementation, $F_{c,0}$ was –0.16 μmol m$^{-2}$ s$^{-1}$, while $F_c$ was 0.13 μmol m$^{-2}$ s$^{-1}$. Within $|QF_{WPL}| \leq 1$, $F_{c,0}$ and $F_c$ were lower by 0.11 μmol m$^{-2}$ s$^{-1}$ and 0.08 μmol m$^{-2}$ s$^{-1}$, respectively, attaining –0.05 μmol m$^{-2}$ s$^{-1}$ and 0.05 μmol m$^{-2}$ s$^{-1}$.


Nonetheless, the need to specially check $CO_2$ fluxes within $|QF_{WPL}| > 1$ does not necessarily imply an error in $CO_2$ flux measurement, and it might suggest further research into validating the application of the WPL correction to small $CO_2$ fluxes, especially on sign-changing $CO_2$ fluxes and fluxes requiring a check based on the $QF_{WPL}$. The verification and





validation process can involve a comparison with $CO_2$ flux measurements obtained using a closed-path gas analyzer, which

is a limitation in this study. It is worth noting that achieving complete isothermal conditions in the measurement volume of the closed-path sensor in the closed-path gas analyzer is essential to obtain the most accurate fluxes at $CO_2$ fluxes < 5 μmol $m^{-2}$ $s^{-1}$, as suggested by Jentzsch et al. (2021).

## 4 Conclusions

A comprehensive analysis of $CO_2$ flux patterns at the tropical coast reveals a dynamic flux characterized by fluctuating

magnitudes and exhibiting periods of $CO_2$ emissions and $CO_2$ uptakes, where the coastal waters predominantly act as a sink. The diel cycle showed fluctuations in the $CO_2$ flux, with smaller magnitudes during the daytime and greater uptake movements during the nighttime.

The application of the Webb-Pearman-Leuning (WPL) correction resulted in changes in the sign of the $CO_2$ flux, indicating a

shift from $CO_2$ sink to $CO_2$ source. Sign changes occur frequently, accounting for over half of the available data, with a substantial number of sign change events during the afternoon hours. The different results obtained from these fluxes can significantly impact the conclusion regarding carbon exchange at the studied location.

The WPL correction parameters, particularly the second and third terms, play crucial roles in $CO_2$ flux sign changes. Higher

vertical wind speed and lower temperature contribute to the second term of the correction for temperature fluctuations. Particularly, the lower molar density of dry air and higher vertical wind speed contribute to the third term related to water vapor fluctuations, which is the major reason for the sign change of the $CO_2$ flux. The diel cycle analysis further reveals the presence of positive values in the third term of the WPL correction throughout the day, with higher values during the daytime that cause more occurrences of sign changes in the $CO_2$ flux.


The analysis of $QF_{WPL}$ highlights that the majority of $CO_2$ flux data falls within the $|QF_{WPL}| > 1$ category. The difference between $|QF_{WPL}| > 1$ and both $|QF_{WPL}| \leq 0.5$ and $0.5 < |QF_{WPL}| \leq 1$ is notably more pronounced during the daytime. $|QF_{WPL}| > 1$ predominantly exhibits sign change fluxes, especially during the day, coinciding with an increase in instances of sign change in $CO_2$ flux. Within $|QF_{WPL}| \leq 1$, most of the data comprises non-sign change fluxes, while occasional sign change

fluxes persist. The implementation of $QF_{WPL}$ within $|QF_{WPL}| \leq 1$ results in lower values of the WPL correction, especially in terms of vertical wind speed as well as the second and third terms of the WPL correction, ultimately leading to a stronger negative $CO_2$ flux within this study location.

Further research may involve $CO_2$ flux measurement using a closed-path gas analyzer for a more comprehensive

investigation and verification of the application of the WPL correction to small $CO_2$ flux in the coastal sea environment.

Specifically, it could focus on assessing the accuracy and reliability of sign-changing $CO_2$ flux, especially fluxes requiring a check based on the quality flagging associated with the WPL correction.

## Data Availability

The study made use of data acquired from the Muka Head Station, located at the Centre for Marine and Coastal Studies of
Universiti Sains Malaysia. This data can be accessed through the following website: http://atmosfera.usm.my/api.html.

## Author Contribution

All authors contributed in the conception and design of the study. MFS initiated the research, devised the methodology, performed data analysis, and produced all the figures for the paper, along with the initial writing. YY contributed to the interpretation of the results and offered valuable insights for the writing and reviewing process of the manuscript. AES,
HMA, and EJJ provided essential resources to facilitate the research, including data collection and curation. The final manuscript was read and approved by all authors.

## Competing Interests

The authors declare that they have no conflict of interest.

## Acknowledgements

We acknowledge that the Malaysian Research University Network Long-Term Research Grant Scheme (MRUN-LRGS) from the Ministry of Education Malaysia, enabled us to conduct this research. Additionally, we express our gratitude towards Elite Scientific Instruments Sdn. Bhd., our industry partner, for their contribution of sensors that allowed us to take accurate measurements.

## Financial Support

This research was funded from the MRUN-LRGS (Malaysian Research University Network Long-Term Research Grant Scheme), which is administered by the Ministry of Education Malaysia (Grant number: 203.PTEKIND.6777006).



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
