# Peer review of "Assessment of the Webb-Pearman-Leuning Correction Method for Estimating CO2 Flux in a Tropical Coastal Sea"

_EGUsphere, 2023_

## Author Comment (AC1)

Response to Reviewers

| | |
|---|---|
| Reviewer #1 | |
| 1 | **General Comments**: *This paper evaluates the Webb-Pearman-Leuning (WPL) correction for eddy covariance (EC) measurements of $CO_2$ fluxes. The WPL correction is an important technique and crucial for the accuracy of flux measurements. The manuscript is in general well organized and presented. My biggest concern of this study, however, is that the results of WPL correction (Figs. 3-7), albeit rather detailed, is solely the difference between fluxes "with" and "without" the WPL correction, which lacks the "ground truth" to be compared with. In other words, even the authors have detailed knowledge of the sign changes, and the dominant terms in WPL correction, there is no way to assess if the WPL correction actually improves the accuracy of $CO_2$ measurements or makes it worse. To justify the fundamental significance and scientific merit of this study, it needs a reliable third-party in-situ or remotely sensed $CO_2$ flux dataset, independent of the EC tower used in this study, to properly "assess" the accuracy and reliability of the WPL corrections. Otherwise, the current study is a sheer sensitivity analysis of the WPL and its dependence on climatological conditions, which can be performed without actual $CO_2$ flux measurements.* |
| | Thank you for the constructive comments and feedback.

Considering the unavailability of an alternative $CO_2$ flux dataset for comparison due to the underreported nature of the site, we have adjusted our research objective for accuracy, as stated in the seventh paragraph of the Introduction (lines 84 - 86):

"The objective of this research is to investigate the sensitivity of the WPL correction method in estimating $CO_2$ fluxes within a tropical coastal environment, particularly focusing on its response to varying climatological conditions."

While we could not utilize another dataset for comparison, our revised objective aims to comprehensively examine the WPL correction method's sensitivity in our specific environmental context. We have thoroughly refined the manuscript to emphasize this adjusted focus. |
| 2 | **Comment 1.1**: *Figure 1: it will be good to have photo(s) of the actual EC tower and/or map of topography at the measurement site.* |
| | **Response 1.1**: Thank you for your suggestion. We have updated Figure 1 with photos of the station, instruments, and a bathymetry map around the site. |

[Figure]

**Figure 1: The location of the Muka Head Station in Penang, Peninsular Malaysia, with an inset displaying the bathymetric map. The station is labeled with a red circle and box, within which the automated weather station is equipped with eddy covariance and Biomet systems.**

| 3 | **Comment 1.2**: *Equation 1: the correction of (kinematic) sensible heat flux term, should it be potential temperature instead of temperature anomaly, though the difference is small at the sea level?* |
|---|---|
|   | **Response 1.2**: Thank you for your comment. Upon the re-evaluating of Equation 1 and referencing Webb et al.'s 1980 paper (https://doi.org/10.1002/qj.49710644707), the correction of the sensible heat flux term use absolute temperature and not potential or anomaly temperature. |
| 4 | **Comment 1.3**: *Section 3.1, the 1st paragraph, how is $F_{c,0}$ computed? Is that the "raw" $CO_2$ flux? It needs to be clearly defined. Also, is it "diel" or "diurnal" cycle?* |
|   | **Response 1.3**: Thank you for your feedback. $F_{c,0}$ represents the raw $CO_2$ flux. We have revised the first paragraph of Section 3.1 (lines 157-159) to explicitly define $F_{c,0}$ as "raw $CO_2$ flux" and $F_c$ as "WPL-corrected $CO_2$ flux."

 "Throughout the sampling time domain, $CO_2$ flux at the study location acted as $CO_2$ uptake, with the average values of the raw $CO_2$ flux (derived from the first term in Eq. (1) as $F_{c,0}$) and the WPL-corrected $CO_2$ flux ($F_c$) are –0.14 and –0.0061 µmol m$^{-2}$ s$^{-1}$, respectively."

 Regarding the "diel" and "diurnal" terminologies, we have amended the second sentence of the first paragraph in Section 3.1 (line 159) for improved clarity by removing the use of those terms: |

| | |
|---|---|
| | "In Fig. 2, the lowest $CO_2$ flux occurred during the daytime, with the flux closing to equilibrium." |
| 5 | **Comment 1.4**: *Figure 2: plots (a) and (b), the region circumscribed by the dashed lines, representing standard errors (standard deviations?), can be shaded for better clarity. Also the measurement uncertainty for the sensible heat flux (Fig. 2c) should also be shown. In addition, as the vapor flux (latent heat) is also included in the WPL correction, it is also recommended to be shown in this figure.* |
| | **Response 1.4**: Thank you for your suggestions. We have made the necessary modifications to Figure 2 (see below). We have also shaded the regions representing standard errors in the figure. Additionally, the uncertainty for sensible and latent heat fluxes has been included, and the graph for latent heat flux has been incorporated into Figure 2.

[Figure]

 **Figure 2: The climatological variation of diel (a) $F_{c,0}$, (b) $F_c$, (c) sensible heat flux, and (d) latent heat flux in 2016. The diel cycle values were averaged over the entire year from January to December 2016.**

 Furthermore, we have added paragraphs discussing the added latent heat flux and the uncertainties for sensible and latent heat fluxes in Section 3.1 (lines 196 - 213), as quoted below: |

"The average uncertainty for sensible heat flux is 0.048 W m$^{-2}$. Between 11:30 LT and 17:30 LT, the uncertainty remained below 0.1 W m$^{-2}$. Subsequently, during the evening and morning hours, the uncertainty fluctuated within the range of 0.1 to 0.3 W m$^{-2}$. Evidently, the uncertainty of sensible heat flux was more substantial between 07:30 LT and 09:30 LT (exceeding 0.3 W m$^{-2}$), especially the uncertainty spike of 0.911 W m$^{-2}$ at 08:00 LT, which coincided with higher uncertainty levels of $CO_2$ flux in the morning.

In Fig. 2d, the latent heat flux ranges from 6.5 to 14.5 W m$^{-2}$, with an average of 10.42 W m$^{-2}$. Peaks in latent heat flux occurred at 08:30 LT (13.66 W m$^{-2}$) and 19:00 LT (14.35 W m$^{-2}$), while lows were observed around 06:30 LT (6.89 W m$^{-2}$) and 20:30 LT (6.69 W m$^{-2}$). During the morning hours (between 05:30 LT and 09:00 LT), there was a noticeable decrease in latent heat flux until 06:30 LT, followed by an increase until 08:30 LT, and then a subsequent decrease. Likewise, a discernible decline in latent heat flux between 19:30 LT and 20:30 LT was followed by an increase until 22:30 LT, indicating markedly greater fluctuations during these periods. Noticeably, higher latent heat flux around 08:30 LT coincided with the peak of sensible heat flux.

The average uncertainty of latent heat flux is 0.25 W m$^{-2}$. Throughout daytime hours (from 09:30 LT to 18:30 LT), the uncertainty remained below 1 W m$^{-2}$. However, during the evening until early morning hours, it exceeded 1 W m$^{-2}$. Similar to the uncertainty of sensible heat flux, the uncertainty of latent heat flux can escalate in the morning, reaching beyond 2 W m$^{-2}$ and even peaking at 3 W m$^{-2}$ around 01:00 LT."

**Comment 1.5**: *Figure 2: the caption states that all plots are "climatological" variation, so it is understood that the diurnal cycle is averaged over the entire year (January to December 2016). This should be clarified and explicitly stated.*

**Response 1.5**: Thank you for your feedback. Based on your suggestion, we have revised the caption of Figure 2 in Section 3.1 (lines 191-192) to explicitly clarify that the climatological variation is the diel cycle values averaged over the entire year from January to December 2016:

"Figure 2: The climatological variation of diel (a) $F_{c,0}$, (b) $F_c$, (c) sensible heat flux, and (d) latent heat flux in 2016. The diel cycle values were averaged over the entire year from January to December 2016."

**Comment 1.6**: *Figure 2d can be grouped with Fig. 3 to show the results and analysis of the discrepancy.*

**Response 1.6**: Thank you for the suggestion. We have grouped Figure 2d with Figure 3 to present the results and analysis of the discrepancy, aiming for a more coherent representation. See below for the new figure.

[Figure]

**Figure 3: (a) The difference value between $F_{c,0}$ and $F_c$, and (b) the quantification of sign change and non-sign change occurrences in a diel cycle.**

---

## Author Comment (AC2)

Response to Reviewers

| | |
|---|---|
| Reviewer #2 | |
| 1 | **General Comments**: *Sigid and others quantify the impact of the WPL correction on eddy covariance measurements using an open path sensor over a coastal sea near the shore. I read the manuscript with interest but feel that unfortunately the framing of the manuscript misses the mark. This is because the WPL correction either should be applied to measured fluxes in open path systems – because it is required to satisfy the mass balance – or it should not in closed path systems with adequate pressure and temperature dampening because not applying it satisfies the mass balance. Studying its impacts serves little purpose because one is studying the consequences of balancing mass or not, which is not of interest.* |
| | Thank you for your concern and comment. |
| | Accurate measurements of $CO_2$ flux in coastal waters are essential for the comprehensive understanding of global carbon processes, which ensures the precision of carbon source and sequestration projections. |
| | We believe investigating the application of the WPL correction over tropical coastal waters is essential and of great interest due to the different environments the tropical coastal seas present, e.g., low wind speeds, high air temperature, humidity, sizeable latent heat influences, etc. These conditions may not only lead to small $CO_2$ fluxes but also affect the values of the terms of the WPL. Previous research on WPL and open-path systems has focused on investigations in open seas with high wind speeds, lower air temperature, and humidity. Therefore, we aim to enumerate the extent of the WPL application for tropical coastal waters. |
| | The WPL correction method was reported to introduce inaccuracies to $CO_2$ flux measurements of small fluxes in the European High Arctic due to the influence of sensible heat fluxes. They found that the correction can substantially affect the actual flux (Jentzsch et al., 2021). Furthermore, our motivation to investigate the WPL correction for these waters is due to the previously observed small $CO_2$ flux values near $0\,\mu\text{mol m}^{-2}\,\text{s}^{-1}$, rooted in the previous work by Yusup et al. (2023). Hence, this prompted the initiation of this study and included even more relevant parameters in the analysis than what was studied in the aforementioned paper (i.e., air temperature, pressure, molar density of water vapor, dry air, and wind speed). |
| | Building on those findings, we hypothesized that the WPL correction might not be accurate or sensitive enough, especially in the small $CO_2$ flux ranges in the proximity of $0\,\mu\text{mol m}^{-2}\,\text{s}^{-1}$ for the tropical coastal waters due to its associated environmental conditions. One notable implication is that the negative $CO_2$ flux can become a positive flux when the WPL correction is applied, which would change the classification of the site from a carbon sink to a carbon source. |

Therefore, we insist that this study is timely and of great interest to researchers who intend to survey the carbon emissions and uptake of the tropical coastal sea using similar methodologies.

| 2 | **Comment 1.1**: *There may be some important technical notes to be made over a warm sea dominated by latent heat fluxes where the WPL terms will be quite small, and it is interesting that they impact CO2 fluxes, but this may be more of a curiosity. Perhaps worthy of very brief mention if only to emphasize the importance of WPL correction for open path systems. (As an aside, I have heard the argument that open path eddy covariance shouldn't be applied over open water, but from the materials presented I was unable to ascertain why.)* |
|---|---|

**Response 1.1**: Thank you for your suggestion. For your information, this was also suggested by reviewer 1. The following describes latent and sensible heat changes and their influences on WPL.

In Fig. 2d, the latent heat flux ranges from 6.5 to 14.5 W m$^{-2}$, with an average of 10.42 ± 0.25 W m$^{-2}$. Similar to the uncertainty of sensible heat flux, the uncertainty of latent heat flux escalates in the morning, reaching beyond 2 W m$^{-2}$. Evidently, the more substantial uncertainty of sensible heat flux between 07:30 LT and 09:30 LT (exceeding 0.3 W m$^{-2}$), especially the uncertainty spike of 0.911 W m$^{-2}$ at 08:00 LT, coincided with higher uncertainty levels of $CO_2$ flux in the morning. Furthermore, peaks in latent heat flux were observed to occur at 08:30 LT (13.66 W m$^{-2}$) and 19:00 LT (14.35 W m$^{-2}$), while lows were observed around 06:30 LT (6.89 W m$^{-2}$) and 20:30 LT (6.69 W m$^{-2}$). Notably, the spike of latent heat flux around 08:30 LT coincided with the peak of sensible heat flux, whereas the dip of latent heat flux around 20:30 LT corresponded to the reduced sensible heat flux.

The difference between $F_c$ and $F_{c,0}$, shown in Fig. 3a, is generally within the range of 0.05–0.2 μmol m$^{-2}$ s$^{-1}$, with the third term of the WPL correction potentially making a substantial contribution, and latent heat flux could be influencing the correction alongside sensible heat flux. The reduced WPL correction values coincided with the notable drop in latent heat flux and lower sensible heat flux around 06:30 LT before the peaks at 08:30 LT and around 20:30 LT. Meanwhile, the rising trend in latent heat flux from 09:00 LT to 19:00 LT corresponded with the increase in the WPL correction value. Notably, the peak of difference value between $F_c$ and $F_{c,0}$ is 0.29 μmol m$^{-2}$ s$^{-1}$ at 08:00 LT, with a substantial increase of 0.2 μmol m$^{-2}$ s$^{-1}$ from the prior time (07:30 LT) and a noticeable decrease of 0.16 μmol m$^{-2}$ s$^{-1}$ at the following time (08:30 LT), coinciding with the spike in latent heat flux and high sensible heat flux during this period.

[Figure]

**Figure 2: The climatological variation of diel (a) $F_{c,0}$, (b) $F_c$, (c) sensible heat flux, and (d) latent heat flux in 2016. The diel cycle values were averaged over the entire year from January to December 2016.**

[Figure]

**Figure 3: (a) The difference value between $F_{c,0}$ and $F_c$, and (b) the quantification of sign change and non-sign change occurrences in a diel cycle.**

| 3 | **Comment 1.2**: *For these reasons the manuscript should be rejected in its present form as it explains – in quite a bit of detail – the consequences of not balancing mass. It misses an enormous opportunity to explain the mechanisms that underlie the observed fluxes, especially the interesting results that CO2 uptake is greater at night (this was unexpected for me, and I am curious to know why), seasonal patterns in flux, the potential influence of different currents and water movements on flux, and how fluxes may or may not be changing over time. Reframing the manuscript to focus on the causes of observations after applying the WPL term would make it interesting and help the community understand this unique system.* |
|---|---|
| | **Response 1.2**: We appreciate your comment and concern.

Our prior work extensively addressed the mechanisms underlying the observed fluxes, i.e., Yusup et al. (2023) and Swesi et al. (2023). The papers included the seasonal patterns analysis. A summary of what was discussed in those papers is below.

Diverse environmental and atmospheric surface layer parameters influence the $CO_2$ exchange between the coastal sea and the atmosphere. According to Yusup et al. (2023), the shift in $CO_2$ flux from functioning as a carbon sink during the night to a carbon source during the day was linked to the differential temperature between the water and air temperatures. A higher (lower) difference in seawater temperature to air temperature tended to support increased $CO_2$ emission (uptake). Under stable conditions, a positive and higher temperature difference resulted in enhanced positive $CO_2$ flux, while the conditions with a lower temperature difference led to heightened negative $CO_2$ flux.

Additionally, research by Yusup et al. (2023) discovered that under stable atmospheric conditions, low wind speeds intensified $CO_2$ flux, while stronger winds resulted in high negative flux during unstable circumstances. The study also highlights the impact of developing waves on $CO_2$ flux in stable atmospheric conditions, contrasting with smoother waves observed during unstable circumstances. As negative flux was noted in developing waves and positive flux exhibited the opposite pattern, surface roughness changes had a more substantial impact on negative flux than positive flux. These underscore the influence of atmospheric stability, winds, and waves on $CO_2$ flux at the study location.

The tropical coastal sea's capability to absorb or release $CO_2$ is also influenced by seasonal changes at the study site, with the Southwest Monsoon acting as a source and the Northeast Monsoon as a sink (Yusup et al., 2023; Swesi et al., 2023). The Southwest Monsoon experienced very unstable atmospheric stability to potentially intensify $CO_2$ emission from the water surface, while the Northeast Monsoon was characterized by weaker unstable circumstances and strong winds (Yusup et al., 2023). Furthermore, according to Swesi et al. (2023), the coastal sea's $CO_2$ source capability during the Southwest Monsoon can be attributed to both low photosynthetically active radiation and concentration of chlorophyll-a, which is the opposite of the high chlorophyll |

concentration during the Northeast Monsoon to cause $CO_2$ uptake. The elevated levels of chlorophyll observed during the Northeast Monsoon may result from upwelling mechanisms and increased nutrient accessibility linked to the rise in wind speed and the decline in water temperature (Swesi et al., 2023).

Responding to the comment, we elaborated below on the difference in the WPL correction values between daytime and nighttime, further clarifying the transition from negative to positive $CO_2$ flux during the daytime due to the WPL correction.

On average, the WPL correction values are not significantly different between daytime and nighttime. The average value of the WPL correction during the daytime is 0.150 $\mu$mol m$^{-2}$ s$^{-1}$, while the average during the nighttime is 0.146 $\mu$mol m$^{-2}$ s$^{-1}$. However, the range of WPL correction values during the daytime (–1.019 to 2.749 $\mu$mol m$^{-2}$ s$^{-1}$) shows a considerable difference compared to the range during the nighttime (–2.552 to 1.322 $\mu$mol m$^{-2}$ s$^{-1}$). The higher positive range values of the WPL correction during the daytime can further explain the more frequent occurrences of a sign change from negative $CO_2$ flux to positive $CO_2$ flux during the daytime.
* * *
| 4 | **Comment 1.3**: *As minor comments I'm not sure why so many wind directions were removed from the analysis; was this due to the impact of the tower? It seemed a bit extreme and perhaps unnecessary to remove so many datapoints. The manuscript is also overly verbose; any word and sentence that isn't necessary to explain key findings should be removed. Focusing the study on science rather than required technical corrections will result in a valuable contribution to the literature.* |
|---|---|
| | **Response 1.3**: Thank you for your feedback. Apart from the poor-quality flags in the recorded measurements, as discussed in Section 2.1, the removal of $CO_2$ fluxes by the wind directions primarily stems from the research focus on fluxes originating from the water surface. We have included this explanation in the same section, as detailed below: |

In this research, $CO_2$ fluxes associated with winds originating from directions >315° and <45° were retained, whereas the fluxes with winds coming from other directions were removed during the data processing. This removal of $CO_2$ fluxes by the wind directions is mainly due to the research focus on fluxes coming from the water surface, and this was based on the standard deviation ratio for the vertical wind speed component and the friction velocity, applicable only to wind directions >315° and <90° (Yusup et al., 2018). Furthermore, wind speed data collected inland from the south to the west of the station (>45° and <315°) were omitted because of the poor-quality flags in the recorded measurements.

Additionally, we have carefully revisited the manuscript to address verbosity. We have made necessary revisions, removing extraneous words and sentences that do not contribute significantly to explaining key findings.

**References**

Jentzsch, K., Boike, J., and Foken, T.: Importance of the Webb, Pearman, and Leuning (WPL) correction for the measurement of small CO2 fluxes, Atmos Meas Tech, 14, 7291–7296, https://doi.org/10.5194/amt-14-7291-2021, 2021.

Swesi, A., Yusup, Y., Ahmad, M. I., Almdhun, H. M., Jamshidi, E. J., Sigid, M. F., Ibrahim, A., and Kayode, J. S.: Seasonal and Yearly Controls of CO2 Fluxes in a Tropical Coastal Ocean, Earth Interact, 27, https://doi.org/10.1175/EI-D-22-0023.1, 2023.

Yusup, Y., Alkarkhi, A. F. M., Kayode, J. S., and Alqaraghuli, W. A. A.: Statistical modeling the effects of microclimate variables on carbon dioxide flux at the tropical coastal ocean in the southern South China Sea, Dynamics of Atmospheres and Oceans, 84, 10–21, https://doi.org/10.1016/j.dynatmoce.2018.08.002, 2018.

Yusup, Y., Swesi, A. E., Sigid, M. F., Almdhun, H. M., and Jamshidi, E. J.: The relationship between carbon dioxide flux and environmental parameters at a tropical coastal sea on different timescales, Mar Pollut Bull, 193, 115106, https://doi.org/10.1016/j.marpolbul.2023.115106, 2023.